# Side Effects of Mixing Vaccines against COVID-19 Infection among Saudi Population

**DOI:** 10.3390/vaccines10040519

**Published:** 2022-03-27

**Authors:** Mohammed Merae Alshahrani, Abdulaziz Alqahtani

**Affiliations:** 1Department of Clinical Laboratory Sciences, Faculty of Applied Medical Sciences, Najran University, Najran 61441, Saudi Arabia; 2Clinical Laboratory Sciences, College of Applied Medical Sciences, King Khalid University, Abha 61321, Saudi Arabia; ayali@kku.edu.sa

**Keywords:** COVID-19, SARS-CoV-2, vaccine, side effects

## Abstract

Background: Mixing two different vaccines has been utilized to minimize the impact of any supply chain interruptions and to combat the COVID-19 pandemic in Saudi Arabia. We conducted this study to evaluate the side effects, if any, associated with the mixed vaccination approach. Methods: An online survey study was administered among COVID-19 vaccine recipients in Saudi Arabia. Symptoms post vaccination were assessed in 311 vaccinated participants with two matched doses of either Oxford–AstraZeneca or Pfizer–BioNTech vaccines, or two mixed doses, respectively. Results: After the second dose, around 31% of the matched vaccine group reported no symptoms, while only 6% of the mixed vaccine group reported no symptoms. Most of the side effects after the second dose associated with matched vaccines were injection site pain (46%), while the mixed vaccines group reported significantly more symptoms compared with the matched vaccine group, which included fever (41%), fatigue (66%), muscle pain (44%), chills (17%) and injection site pain (60%). Conclusion: The data suggest the overall safety of the mixed vaccination protocol; however, it might be associated with side effects such as fever, fatigue, muscle pain, chills, and injection site pain. Further studies with a larger cohort size could shed more light on this aspect, which would be imperative for deciding to utilize a mixed vaccination approach.

## 1. Introduction

A new member of the human coronavirus family, severe acute respiratory syndrome coronavirus 2 (SARS-CoV-2), was first recognized in an pandemic of the highly infectious respiratory illness in Wuhan, China, in 2019. It is known that the virus causes coronavirus illness (COVID-19), which is asymptomatic or moderate in the majority of patients; nevertheless, 20% of infected people are at risk of developing severe to life-threatening conditions [1]. Coronavirus caused a global pandemic, which has considerably impacted Saudi Arabia and other countries. Noticeably, Saudi health authorities took extraordinary and cautious preventative steps, which included barring foreign flights, shutting mosques, schools, colleges, and locking down the kingdom completely to lessen the disease’s impact [2]. Along with restricting the spread of virus via avoiding person-to-person contact, effective strategies to control the disease include the development of a vaccine with further mass vaccination of the population and the development of anti-viral drugs [3]. Keeping this aspect in view and understanding the global crisis caused by the virus, researchers around the world have sped up the vaccine development process and produced a few vaccines showing efficacy against SARS-CoV-2 [4,5]. Many SARS-CoV-2 vaccines were made available for emergency use from December 2020, including BNT162b2 (Pfizer–BioNTech), ChAdOx1 (AstraZeneca), mRNA-1273 (Moderna), Ad26.COV2.S (Johnson & Johnson/Janssen) and Sinopharm and Sinovac [3]. On the 10 December 2020 and 18 February 2021, Saudi Arabia also authorized the use of BNT162b2, as well as ChAdOx1, for vaccination against COVID-19 in the Saudi population. The efficacy and safety of both these vaccines have been proved by various clinical studies [6,7]. The BNT162b2 vaccine, which is based on mRNA technology, and the ChAdOx1 AstraZeneca vaccine, which uses chimpanzee adenovirus (ChAdOx1) as a viral vector for the expression of the SARS-CoV-2 spike protein, have demonstrated potential efficacy against SARS-CoV-2 infection. Studies have reported vaccine efficacy in terms of relative risk reduction by the vaccine, which is based on the ratio of the attack rate with and without vaccine. Based on the ranking by the reported efficacy of the vaccines, BNT162b2 showed a relative risk reduction of 95%, and a 67% risk reduction was reported for the ChAdOx1 AstraZeneca vaccine [8,9]. A variety of mild to tolerable adverse effects, including fatigue, headaches, chills, muscle and joint pains, fever, and injection site pain, have been linked to both vaccines in clinical studies [8,9]. Relative risk reduction, on the other hand, has to be assessed against the background risk of catching COVID-19 infection, which varies considerably between populations and periods. Although the relative risk reduction takes into account just those who potentially benefit from the vaccination, the absolute risk reduction, which is the difference between attack rates with and without a vaccine, takes into account the whole population. Absolute risk reduction is often neglected since it has a considerably smaller impact size than relative risk reduction: 1.3% for the Oxford–AstraZeneca and 0.84% for the Pfizer–BioNTech vaccines [10].

However, this vaccination programme has been related to various difficulties, as is evident from published data. Various adverse effects have led to the suspension of the use of a few COVID-19 vaccines in some countries. Moreover, countries in impoverished areas have had difficulty giving adequate vaccination doses to their populations. As a result, it was thought that the flexibility to mix and match vaccines would speed the immunization process and reduce the effect of any supply chain delays, allowing vaccination programs to be more flexible in the future. This strategy, which includes using heterologous vaccines in the first and second doses, might help in overcoming the abovementioned difficulty with a shortage of vaccines, which could largely affect the mass-vaccination programme against COVID-19 [11].

Interestingly, the most important aspect of using the mixed vaccination approach, i.e., studying the side effects post matching or mixing of COVID-19 vaccines, if any, has not yet been elucidated clearly. Few available studies have shown controversial outcomes in this regard. According to a Spanish study, 448 patients were given Oxford–AstraZeneca and Pfizer BioNTech as the first and second doses, respectively, which resulted in poor side effects [12]. Another study compared the side effects caused by a mixed as well as a matched vaccination strategy and found that the side effects of two mixed doses of different vaccines were not worse than the side effects of two matched doses of the same vaccine [13]. However, the Com-COV investigation showed that combining two different vaccines may have greater adverse effects than administering two doses of the same vaccine [14]. Furthermore, a study was performed on 830 people, in which Oxford–AstraZeneca as a prime dose and Pfizer–BioNTech as a boost dose were used, respectively. The occurrence of fever 48 h post vaccination was the most prevalent symptom in the heterologous prime-boost approach compared with other homologous counterparts with either vaccine [15].

We conducted this study to compare the short-term side effects caused by the mixed vaccination approach and matched vaccine approach in a cohort of vaccinated people, based on an online survey using a questionnaire. In our study, participants aged 18 and above in Saudi Arabia received matched or mixed doses with either Pfizer–BioNTech mRNA (BNT162b2) or Oxford–AstraZeneca (ChAdOx1 nCoV-19) vaccines.

## 2. Materials and Methods

Both Pfizer–BioNTech and the Oxford–AstraZeneca vaccines were used for the mass immunization campaign against COVID-19 in Saudi Arabia in early 2021. This four-month retrospective cross-sectional online convenience sampling survey was performed in Saudi Arabia between September and December 2021. Google Forms was used to create a bilingual (Arabic and English) online survey which was sent to the participants through social media. An electronic email was also set up and utilized to assist the communication between the research investigators and participants for any needed inquiry. There were two aspects of this survey. The first part of the questionnaire was meant to gather personal information such as gender, age, chronic health conditions, and SARS-CoV-2 infection history. Vaccination-related data, such as the type and time of the COVID-19 vaccine, including the first and second dose, side effects post COVID-19 vaccine, and the duration of adverse effects, were covered in the second part of the survey questionnaire.

Furthermore, we also included a section in the survey questionnaire for reporting any other unlisted side effects that our study participants might have experienced in addition to the most common side effects that have been reported in other studies, such as pain at the injection sites, tiredness, headache, fever, chills, dizziness, and vomiting [8,9]. The ethical approval committee at Najran University authorized this research and written informed consent was obtained from all participants. This study included participants who were given either matched or mixed double doses of Pfizer–BioNTech or Oxford–AstraZeneca vaccines in Saudi Arabia. We excluded all the individuals who declined to participate in this survey study and those who were not immunized against COVID-19 or given a vaccine, apart from the Pfizer–BioNTech or the Oxford–AstraZeneca vaccines. Before applying the chi-square test in its classical form, we tested the hypothesis to determine whether two categorical or nominal variables are likely to be related or not by using the Pearson chi-square test of independence. In the first step, we calculated the sum of each row, and the sum of each column of side effects for both matched and mixed groups. In the second step, the expected values were then calculated for each cell of an observed value. Third, the Pearson chi-square value was calculated alongside with the degrees of freedom (df) value to obtain the significance using the chi-square function. The significance of the Pearson chi-square value (28.93) with 9 df value was *p* = 0.00066. Bivariate analysis was performed using the chi-square test. Binary logistic regression was used to assess how the type of vaccine can predict the odds of having each side effect. For the above regression, odd ratio (OR) and the respective 95% confidence interval (CI) were then estimated. The assumptions of the chi-square test and logistic regression were explored; no violations were observed [16]. We conducted a statistical analysis of the data using SPSS v.23 (IBM Corp, Armonk, NY, USA), with *p* ≤ 0.05 as significant.

## 3. Results

### 3.1. Study Participants’ General Characteristics

A total of 311 participants were included in this study. Out of the 311 participants, 147 (47.2%) were females and 164 (52.7%) were male (Table 1). The participants’ age ranged between 12 and 65 years, with a median age of 38.5 (Table 1). Most of the participants were physically healthy; however, some of the respondents (20.2%) reported chronic health conditions including but not limited to asthma, heart diseases, food and skin allergy, hypertension, and others. The details of the reported chronic health conditions are mentioned in Table 1. All 311 participants were vaccinated with two doses of either Pfizer–BioNTech or Oxford–AstraZeneca vaccines against COVID-19 infection.

Furthermore, the study participants were vaccinated with two doses of the same vaccine or received two different doses of the vaccines mentioned above. Based on the type of vaccines our participants received, we categorized them into two groups; those who were vaccinated with one type of vaccine for the two doses (*n* = 237, 76.2%), i.e., received two doses of either the Pfizer–BioNTech or Oxford–AstraZeneca COVID-19 vaccines, and those who were vaccinated with different vaccines (*n* = 74, 23.7%), i.e., received one dose of Pfizer–BioNTech followed by a second dose of Oxford–AstraZeneca and vice versa (Table 1).

### 3.2. Mixed and Matched COVID-19 Vaccines Provide Protection against SARS-CoV-2

As shown in Figure 1A, only 29.6% (*n* = 92) of participants in both mixed and matched groups were infected with SARS-CoV-2. Interestingly, most of the infected cases (*n* = 57, 62%) reported infection before receiving the COVID-19 vaccine, while only 31.5% (*n* = 29) were infected with COVID-19 infection after the first dose of either Pfizer–BioNTech or Oxford–AstraZeneca vaccines (*p* = 0.001) (Figure 1B). Additionally, fewer COVID-19 cases (*n* = 6, 6.5%) were reported after receiving the two doses of the vaccines compared with infected cases before vaccination (*p*= < 0.001) (Figure 1B). Furthermore, there was a significant reduction in the incidence of COVID-19 infections among the participants who received the second dose compared with those who received the first dose (*p*= < 0.001) (Figure 1B). These data indicate the potential of COVID-19 vaccines in preventing SARS-CoV-2 infection. Next, we attempted to assess the efficacy of a mixed vaccine strategy against SARS-CoV-2. Unfortunately, all the participants who had COVID-19 after being vaccinated with the two doses were from the matched group, i.e., either received two doses of Oxford–AstraZeneca vaccine or two doses of Pfizer–BioNTech COVID-19 vaccine.

### 3.3. Side Effects of Mixed Vaccination Strategy Compared with Matched Vaccination Efficacy

To evaluate the safety of the mixed vaccination strategy, we asked the participants in mixed and matched groups to report symptoms following the second dose of COVID-19 vaccines. As shown in Table 2, only 6.7% of the respondents in the mixed group had no symptoms after the second dose compared with nearly 31.6% in the matched group (*p*= < 0.001). The rest of the participants in both groups experienced comparable mild symptoms, including headache, diarrhoea, and backache (Table 2). These symptoms were comparably present among respondents in the two groups, with no statistically significant differences. However, we observed that fever, following the second dose, was more prevalent in the mixed group compared with the matched group (41.8% vs. 25.7%; *p*-value 0.008). Fatigue and muscle pain were also more prevalent in the mixed group compared with the matched group, in which fatigue was 66.2% vs. 36.7% (*p*-value 0.001) and muscle pain was 44.5% vs. 28.2% (*p*-value 0.009) (Table 2). Furthermore, we also observed that chills and injection site pain were more prevalent in the mixed group compared with the matched one. Chills was 17.5% vs. 6.7% (*p*-value 0.005), while injection site pain was 60.8%vs. 46.8% (*p*-value 0.03) (Table 2). Additionally, the duration of symptoms, up to 48 h, was longer among the participants in the mixed group (44.5%) than those of the matched group (27.8%) (*p* = 0.007) (Table 2), This suggested that the mixed vaccination strategy might cause longer side effects. Interestingly, there was a significant drop in the duration of symptoms after 48 h (27.8%) compared with 24 h (45.9%) (*p*-value 0.001) in those who were inoculated with matched doses of COVID-19 vaccine (Table 2). Moreover, there was a difference in lasting symptoms after 48 h (44.5%) compared with 24 h (37.8%) in those who were inoculated with mixed doses of COVID-19 vaccine. However, this difference was not statistically significant (*p* = 0.52) (Table 2).

## 4. Discussion

Vaccination is the most effective strategy to control the ongoing COVID-19 pandemic. Several manufacturers had developed COVID-19 vaccines, including, but not limited to, mRNA-based vaccines (Pfizer–BioNTech and Moderna) and viral vector-based vaccines (Oxford–AstraZeneca, Johnson & Johnson, and gam-COVID-Vac) [11]. The majority of COVID-19 vaccines were shown to be safe and effective against SARS-CoV-2 [8,13,14,17]. Due to the globally high demand for COVID-19 vaccines, several countries have approved the administration of using two different vaccines for the prime and boost doses (World Health Organization, https://www.who.int/, accessed on 10 March 2022). However, this mixed vaccination strategy is not well understood in terms of the efficacy and, most importantly, the safety. Therefore, in this present study, we conducted an online survey to investigate the side effects associated with the mixed vaccination approach.

Our analysis showed that COVID-19 vaccines possess potent efficacy in preventing SARS-CoV-2, the causative agent of COVID-19 (Figure 1B). One dose of either Oxford–AstraZeneca or Pfizer–BioNTech decreased the number of COVID-19 cases by nearly half, while almost no cases were reported after receiving two doses of the same vaccine (Pfizer–Pfizer or AstraZeneca–AstraZeneca) (Figure 1B). After completing the two doses in the case of the mixed vaccination approach (Pfizer–BioNTech and Oxford–AstraZeneca), none of the participants had a COVID-19 infection. These results suggest the high efficacy of both the vaccination approaches against SARS-CoV-2. Previous studies reported similar findings regarding the effectiveness of mix-and-match COVID-19 vaccines [12,13,14]. However, the probable reason for there being no COVID-19 cases in the recipients of mixed vaccination was the small number of participants in this group compared with the matched vaccine group (Table 1). Therefore, we could not compare the efficacy of both these approaches.

Our study found that the mixed vaccination approach was associated with more side effects in comparison to the matched vaccination approach. A significant number of participants who received mixed vaccine doses experienced symptoms including fever, fatigue, muscle pain, chills, and injection site pain following the second dose of the COVID-19 vaccine (Table 2). There was an increase in other side effects in participants who received mixed vaccines compared with those who received matched vaccines; however, this difference was not statistically significant (Table 2). However, any strong conclusive statement could not be provided due to small sample size of participants who consecutively received two different vaccines for their prime and booster dose. Symptoms of adverse events lasted longer in participants who consecutively received two different vaccines for the prime and booster dose as compared with participants who received the same vaccine for both doses (Table 2). It is not clear why participants in the mixed vaccination group experienced not only increased reactogenicity, but also had symptoms lasting longer than 48 h, compared with those who received the same type of vaccine for the first and second dose (Table 2). Our findings are in line with the Com-COV study, in which the research team evaluated the efficacy and reactogenicity of the heterologous vaccine regimen ChAdOx1 NCoV-19 (ChAd) COVID-19 vaccine (Vaxzevria, AstraZeneca) and BNT162b2 (BNT) COVID-19 vaccine [15]. With respect to the reactogenicity, the study found that participants who received different types of vaccines for the prime and booster doses had greater systemic reactogenicity [15]. The study also found that fever was more prevalent among participants in the mixed vaccine group, which was reported by nearly 34% of the participants compared with only 10% of participants who received the same type of vaccine for the first and second dose [15].

In contrast, few recent studies reported that mix-and-match vaccine regimens caused similar reactogenicity [12,13,18]. Based on our and previous studies, it is possible that mixing different COVID-19 vaccines may increase systemic reactogenicity. However, more investigations are required to evaluate the side effects of mixed vaccine regimens.

This study is the first study in Saudi Arabia to assess the side effects associated with mixed vaccine regimens, to the best of our knowledge. However, the present study has several limitations. Because of the COVID-19 pandemic and the suggestion to maintain social distancing and preventative measures in Saudi Arabia, we opted to conduct this convenience sampling survey as a web-based study to protect the safety of all study participants. The data were collected through a self-administered online questionnaire, and this might result in a reporting bias. Moreover, community-based surveys would be difficult to perform during this pandemic. Therefore, data collection was performed online as a self-reported survey, and the dissemination of this survey was dependent on the authors’ networks. Furthermore, our convenience sampling may limit the possibility to generalize our findings to a wider population; however, our target sample was hard to reach, because the only way to reach participants was through social media, and therefore, we were not able to apply random sampling in this study. To the best of our knowledge, convenience sampling does not have any restrictions with regard to the statistical tools to be used. Therefore, in our study, the chi-square test was used [16] to check whether the differences between groups were statistically significant or not.

On the other hand, the sample size of the two groups (mixed and matched) was not comparable. We only included 74 respondents in the mixed vaccine group, while 237 respondents were in the matched group, which was because most people in Saudi Arabia received homologous vaccine regimens for both the prime and boost dose (https://www.moh.gov.sa/, accessed on 10 March 2022). Therefore, it was challenging to include more participants in the mixed vaccine group.

Moreover, whether our subjects used pain medications such as ibuprofen, antihistamine, and aspirin followed COVID-19 vaccinations is unknown. Therefore, this could have influenced the reported side effects. For example, those who received these medications might have experience mild or no symptoms compared with the individuals who did not. Since most of our participants did not remember whether they had pain relievers, we could not include this in our analysis. The other limitation is that all of the reported chronic health conditions were among recipients of homologous vaccines, while none of the recipients of heterologous vaccines reported any chronic health condition. Lastly, the interval between the first and second dose in the two groups (mixed and matched groups) was not included in our analysis. Because the recommended intervals between doses of Pfizer–BioNTech and Oxford–AstraZeneca are different, being 2 and 6 weeks, respectively, and because of the observed inconsistent interval between the first and second dose among our study participants, we excluded it from our study. Given this limitation, further research is needed to address these issues. It would be of interest to introduce a clinical trial model evaluating the side effects between a group receiving matched doses and a group receiving mixed doses of vaccines against COVID-19 infection. This model might help to further elucidate the mechanism by which mixing and matching vaccine doses might cause side effects, and also to avoid any possible frauds in the completion of formulary online surveys.

## 5. Conclusions

Our study shed light on the practicality of using two different vaccines for prime and booster doses to combat the ongoing COVID-19 pandemic. In this study, we evaluated the short-term side effects of mixing COVID-19 vaccines. We found that participants who received two different COVID-19 vaccines reported significantly more symptoms after the second dose compared with matched vaccine group, which included fever, fatigue, muscle pain, chills, and injection site pain. However, any strong conclusive statement could not be provided due to the small sample size of participants who consecutively received two different vaccines for their prime and booster dose. Therefore, a larger population study is needed to assess the side effects of mixing two different vaccines against SARS-CoV-2 infection.

## Figures and Tables

**Figure 1 vaccines-10-00519-f001:**
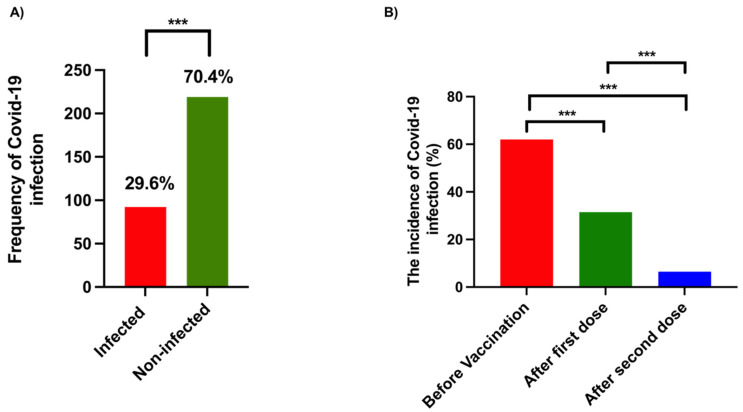
The frequency and incidence of COVID-19 infection within this study. (**A**) The total number of participants was 311 people, in which the frequency of infected people with COVID-19 was 92 (29.6%), and non-infected people was 219 (70.4) before the vaccination system started in Saudi Arabia. (**B**) The incidence of COVID-19 infection among participants in this study, regarding the frequency of infected people before the vaccination system, was 57 (62%), while after the first dose it was 29 (31.5%) and after the second dose it was 6 (6.5%). Statistical analysis was performed using the chi-square test. *** *p* < 0.001.

**Table 1 vaccines-10-00519-t001:** Gender, age, and risk factors for COVID-19 infection among participants within this study. This table shows the matched and mixed vaccines, including the first and second doses with either Pfizer–BioNTech or Oxford–AstraZeneca. Data are shown as numbers (percentages). The chi-square test was used to check whether the differences between groups are statistically significant or not. *p*-value of the chi-square test is presented against each group that either matched or mixed vaccines against COVID-19 infection.

Variable	Matched Vaccines, *n* = 237 (76.2%)	Mixed Vaccines, *n* = 74 (23.7%)	Total, *n* = 311 (100%)	*p*-Value
Sex, *n* (%)				
Male	117 (49.3%)	47 (63.5%)	164 (52.7%)	0.18
Female	120 (50.6%)	27 (36.4%)	147 (47.2%)	0.13
Age, *n* (%)				
12 to 18	40 (16.8%)	5 (6.7%)	45 (15.5%)	0.03
18 to 30	62 (26.1%)	32 (43.2%)	94 (30.2%)	0.04
30 to 45	115 (48.5%)	34 (45.9%)	149 (47.9%)	0.75
45 to 65	20 (8.4%)	3 (4%)	23 (7.4%)	0.24
The presence of health conditions, *n* (%)				
Diabetes	6 (2.5%)			
Hypertension	9 (3.7%)			
Food and skin allergy	24 (10.1%)			
Hypothyroidism	1 (0.4%)			
Rheumatoid	1 (0.4%)			
Asthma	18 (7.5%)	0	63 (20.2%)	N/A
Kidney diseases	1 (0.4%)			
Sickle cell anaemia	1 (0.4%)			
Heart diseases	2 (0.8%)			

**Table 2 vaccines-10-00519-t002:** The adverse effects after the second dose between participating groups who matched or mixed COVID-19 vaccines. The table illustrates the adverse effects after the second dose between groups who matched or mixed COVID-19 vaccines, including the BNT162b2 (Pfizer–BioNTech) and the ChAdOx1 (AstraZeneca) vaccines. The table shows the frequency of no symptoms after the second dose between participants who matched or mixed the first and the second dose with either BNT162b2 (Pfizer–BioNTech) or ChAdOx1 (AstraZeneca) vaccines, respectively. Furthermore, the table shows the duration of symptoms in participating groups after the second dose between matched and mixed vaccines. Statistical analysis was performed using the chi-square test. * *p* < 0.05, ** *p* < 0.01, *** *p* < 0.001, ns = not significant. ^a^ Derived from chi-square test. ^b^ Level of significance (alpha) = 0.05. h = hours.

Side Effect	Matched *n* = 237 (76.2%)	Mixed *n* = 74 (23.7%)	*p*-Value ^a,b^	OR	95% CI
Fever	61 (25.7%)	31 (41.8%)	0.008 **	0.614	0.370–1.018
Fatigue	87 (36.7%)	49 (66.2%)	<0.001 ***	0.554	0.317–0.966
Headache	72 (30.3%)	28 (37.8%)	0.23 ns	0.802	0.458–1.404
Muscle pain	67 (28.2%)	33 (44.5%)	0.009 **	0.633	0.346–1.159
Chills	16 (6.7%)	13 (17.5%)	0.005 **	0.384	0.165–0.892
Diarrhoea	12 (5%)	5 (6.7%)	0.57 ns	0.749	0.209–2.680
Injection site pain	111 (46.8%)	45 (60.8%)	0.03 *	0.770	0.256–2.312
Backache	32 (13.5%)	16 (21.6%)	0.09 ns	0.624	0.312–1.248
Duration of symptoms					
0 h (No symptoms)	75 (31.6%)	5 (6.7%)	<0.001 ***	4.683	1.728–12.69
24 h	109 (45.9%)	28 (37.8%)	0.21 ns	1.215	0.744–1.985
48 h	66 (27.8%)	33 (44.5%)	0.007 **	0.624	0.346–1.125

## Data Availability

The data presented in this study are available on request from the corresponding author.

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
