# Peer review of "Side Effects of Mixing Vaccines against COVID-19 Infection among Saudi Population"

_vaccines, 2022, doi:10.3390/vaccines10040519_

Round 1

Reviewer 1 Report

It will be of interest if the authors spent more time and concentrate  on development of the more balanced study.
 I strongly recommend to update & correct all issue from Major Comments 1-8.  
Sincerely, Referee

Major comments:

Major Comment 1: Sample size is extremely small.

Major Comment 2: There is no appropriate clinical trial model  introduced, no discussion on tuning on the parameters-

Major Comment 3: Referee disagree that there is no antiviral therapy. This statement needs justification by reference. 

Major Comment 4: line 58: 95% effectiveness should be carefully checked, see doi: 10.1016/S2666-5247(21)00069-0

Major Comment 5: line 74: how much the results from mice are projectile to humans? What about incompatibility of protein reactions?

Major Comment 6: line 83: arbitrary large   immunoglobulin G (IgG) and IgA anti-spike (S) response does not necessary guarantee better immunity reaction, too high values are not optimal.

Major Comment 7: based on what you explain about your online study, it can be called maximally non systematic observational study, but what is the information quality of such questionnaires?

Major Comment 8: Application of chi^2 test is very disputable, since sample is not at random.   One of the assumptions of the Chi-square is that sampling method is  simple  random sampling. thus using of p-value cut-off 0.05 does not holds its nominal significance level.  Statistics should be improved, chi^2 test in its classical form is not reliable.
Thus, several conclusions are unjustified from the obtained p-values. 

Author Response

Dear Editor,

We are submitting our revised manuscript entitled “Side Effects of Mixing Vaccines Against COVID-19 Infection Among a Random Saudi Population” to Vaccines for evaluation for publication. Reference number for a previous version of this manuscript is 1610519.

Thank you for giving me the opportunity to submit a revised draft of my manuscript to “Vaccines journal”.

I appreciate the time and effort that you and the reviewers have dedicated to providing your valuable feedback on my manuscript. I am grateful to the reviewers for their insightful comments on my paper. I have been able to incorporate changes to reflect most of the suggestions provided by the reviewers.

Here is a point-by-point response to the reviewers’ comments and concerns. 

 Sincerely,

Mohammed Merae Alshahrani, PhD.

Assistant Professor

Najran University, Saudi Arabia  

Point-by-Point Response to Comments: (Vaccines-1610519):

Reviewer 1

  • Major Comment 1: Sample size is extremely small:

The authors thank the reviewer for the comments. We agree with the reviewer that the sample size is small. We included a total of 311 participants; 237 received matched vaccines while only 74 received mixed vaccines. The majority of people in Saudi Arabia had homologous vaccines. People were concerned and worried about potential side effects or consequences having two different doses as there are not enough studies on that. For this reason, it was very difficult to include more participants in the mix-vaccine group. Therefore, the size of our sample was mentioned as a limitation in the revised version of the manuscript (line 560-564).    

  • Major Comment 2: There is no appropriate clinical trial model introduced, no discussion on tuning on the parameters-

The authors thank the reviewer for this comment. Since we used an online-based survey questionnaire to collect our data and because there was no clinical trial involved in our study, we did not introduce a clinical trial model. However, we included preliminary data from safety and efficacy trials in the manuscript (line 165-174). So, we wish to clarify to the learned reviewer that this was a cross-sectional prospective study based on the data collected from random participants who had been previously vaccinated. We did not vaccinate the participants for the both the study groups. This was not a clinical trial.

  • Major Comment 3: Referee disagree that there is no antiviral therapy. This statement needs justification by reference. 

The authors thank the reviewer for this comment. Most of the anti-viral therapies used against COVID-19 were re-purposed drugs which have been used previously for same family of viruses. Therefore, the statement provided by the authors was meant that there are no effective therapy (Mayo Clinic) especially against COVID-19 caused by the omicron variant. Despite these drugs are promising, but they also have limitations. Therefore, with respect to the reviewer for this comment, we decided to remove the sentence to reduce the discussion around this point as the aim of our study is focusing on the side effects of mixing vaccines against COVID-19 compared to matching ones.

  • Major Comment 4: line 58: 95% effectiveness should be carefully checked, see doi: 10.1016/S2666-5247(21)00069-0

We are thankful to the learned reviewer for raising this point. It was an error in expression of the information. We worked on this point raised by the learned reviewer and have amended the sentence accordingly to avoid any controversial statement regarding data (line 168-172).

  • Major Comment 5: line 74: how much the results from mice are projectile to humans? What about incompatibility of protein reactions?

We totally agree with the learned reviewers that data of mice should be assessed for similar projection in the humans. However, the rationale of our study was not to study the various aspects of efficacy of COVID-19 vaccines undertaken in the study and assess the incompatibility related immunological reactions, but this was a retrospective cross-sectional online survey study with a aim to compare the side effects of two vaccination strategies viz. Mixed vaccine approach and use of matched vaccine for both the doses in a cohort of population. We also agree that results from mice are not 100% projectile to humans and need for clinical trial assessment of heterologous prime-boost regimens. Therefore, we included in the manuscript (line 245-255) clinical trial studies that showed that mixed doses of vaccines elicited greater immune responses than matched vaccines did.  

  • Major Comment 6: line 83: arbitrary large immunoglobulin G (IgG) and IgA anti-spike (S) response does not necessary guarantee better immunity reaction, too high values are not optimal.

 Thank you very much for pointing this important information out. The study showed that two doses of AstraZeneca and Pfizer vaccines induced not only a higher humoral immune response but also a cellular immune response, which plays a critical immunity role. Therefore, a combination of strong humoral and cellular responses should result in a better immunity reaction. We also added this point in the revised manuscript (line 249-252).

  • Major Comment 7: based on what you explain about your online study, it can be called maximally nonsystematic observational study, but what is the information quality of such questionnaires?

Thank you for this comment. Although a survey questionnaire is a convenient and quick method to easily collect data, it could result in a reporting bias and participants may have difficulties precisely reporting their adverse effects. This was also added as one of the limitations of our study in the revised manuscript (line 552-558).

  • Major Comment 8: Application of chi^2 test is very disputable, since sample is not at random. One of the assumptions of the Chi-square is that sampling method is simple random sampling. thus, using of p-value cut-off 0.05 does not holds its nominal significance level.  Statistics should be improved, chi^2 test in its classical form is not reliable. Thus, several conclusions are unjustified from the obtained p-values.

The authors thank the learned reviewer for this valuable comment. As far as we know that random sampling includes ease of use and accuracy of representation, we do believe that no easier method exists to extract a research sample from a larger population than simple random sampling. Furthermore, although simple random sampling is intended to be an unbiased approach to surveying, however, sample selection bias might be occurred. Therefore, we added this in limitations section in the revised manuscript (line 552-558) and we mentioned that also in the materials and methods section that we used random sample survey (line 295).

Reviewer 2 Report

This study compared the rates of short-term adverse effects between 237 individuals who received matched vaccines (Pfizer-BioNTech only or Oxford-AstraZeneca only) and 74 individuals who received mixed vaccines (Pfizer-BioNTech + Oxford-AstraZeneca only). The authors concluded that mixed vaccines resulted in less adverse effects than matched vaccines.

The first major flaw of this study is the small number of participants. Only 74 participants received mixed vaccines. The small number of participants limited the possibility to conclude the difference in adverse effect of two vaccine injection strategies.

The second major flaw of this study is the method of assessing adverse effects of vaccines. This study used online self-reported questionnaire survey. Although online survey is a good method to collect data during the COVID-19 pandemic, using this method to survey adverse effects of vaccines is incomplete. Participants may have difficulties in definite the adverse effects.

The analysis strategy did not follow the aims of this study. Examining the frequency and incidence of COVID-19 infection is not the aim of this study.

English writing of this manuscript needs a good sharp edit. I am also confused about the term “Mixing Vaccination System” in the title.

Author Response

Dear Editor,

We are submitting our revised manuscript entitled “Side Effects of Mixing Vaccines Against COVID-19 Infection Among a Random Saudi Population” to Vaccines for evaluation for publication. Reference number for a previous version of this manuscript is 1610519.

Thank you for giving me the opportunity to submit a revised draft of my manuscript to “Vaccines journal”.

I appreciate the time and effort that you and the reviewers have dedicated to providing your valuable feedback on my manuscript. I am grateful to the reviewers for their insightful comments on my paper. I have been able to incorporate changes to reflect most of the suggestions provided by the reviewers.

Here is a point-by-point response to the reviewers’ comments and concerns. 

 Sincerely,

Mohammed Merae Alshahrani, PhD.

Assistant Professor

Najran University, Saudi Arabia  

Point-by-Point Response to Comments: (Vaccines-1610519):

Reviewer 2

1) This study compared the rates of short-term adverse effects between 237 individuals who received matched vaccines (Pfizer-BioNTech only or Oxford-AstraZeneca only) and 74 individuals who received mixed vaccines (Pfizer-BioNTech + Oxford-AstraZeneca only). The authors concluded that mixed vaccines resulted in less adverse effects than matched vaccines.

The authors thank the reviewer for the comments. We wish to convey to the learned reviewers is that our data suggested that 31% of the participants who received two matched vaccine doses reported no symptoms after the second dose. On the contrary, of the total participants in the other group, who received two different vaccine doses, only 6% reported no adverse events after the second dose. This suggests that more adverse events were associated with the mixed vaccination approach. Moreover, the adverse events were significantly more severe including fever, chills and fatigue in case of mixed approach when compared to matched vaccine approach in which only side effect reported was pain at the injection site.

2) The first major flaw of this study is the small number of participants. Only 74 participants received mixed vaccines. The small number of participants limited the possibility to conclude the difference in adverse effect of two vaccine injection strategies.

The authors thank the reviewer for this comment. We agree that the sample size used in our study was small to reach into a generalized conclusion. However, this was the total number of respondents we were able to include in this study. This was one of the study limitations and this was mentioned in the revised manuscript (Line 560-564).

3) The second major flaw of this study is the method of assessing adverse effects of vaccines. This study used online self-reported questionnaire survey. Although online survey is a good method to collect data during the COVID-19 pandemic, using this method to survey adverse effects of vaccines is incomplete. Participants may have difficulties in definite the adverse effects.

Thank you for this valuable comment. We agree with the reviewer that online self-questionnaire survey could result in a reporting bias and participants may have difficulties precisely reporting their adverse effects. Due to the ongoing COVID-19 pandemic and the preventive measures in Saudi Arabia, we preferred to conduct this study as a web-based study to ensure the safety of our study participants. Therefore, this was the only way to collect data during the pandemic with a lot of restrictions were applied in Saudi Arabia during pandemic. This was added as one of the limitations of our study in the revised manuscript (line 552-558).

4) The analysis strategy did not follow the aims of this study. Examining the frequency and incidence of COVID-19 infection is not the aim of this study.

The authors thank the reviewer for this comment. The aim of the study was to compare the side effect of mixing two different vaccines against Covid-19 infection to matching ones. So, we just added the frequency and incidence to show that vaccination against Covid-19 is decreasing the incidence.

5) English writing of this manuscript needs a good sharp edit. I am also confused about the term “Mixing Vaccination System” in the title.

The authors are grateful to the reviewer for this comment on our paper. The revised manuscript went through extensive English language editing and they are all marked by tracking changes. Furthermore, the title of the manuscript has been changed in the revised manuscript into ‘’Side Effects of Mixing Vaccines Against COVID-19 Infection Among a Random Saudi Population’’.

Round 2

Reviewer 1 Report

Authors tried to answer my major comments, but much more work is needed. Practically, they succeeded to reply completely to Major  comment 3.
The other comments needs more attention and work, according to the following queries, which can help Authors to better address the original Major comments.

    Addendum to    Major Comment 1: Sample size is extremely small:
It should be also clearly written in 5. Conclusion, so you cannot (based of very low sample size and non-comparable sample sizes) guarantee
that  a mix-vaccine strategy is safe. Moreover there is no objective justification of the sentence "Therefore, based on our current study, a mix-vaccine strategy is an effective way  
to compromise the shortages of COVID-19 vaccines and to accelerate the vaccination pro- 296
cess globally." and it should be removed. You should be careful that non-reporting effects these types of survey, and your statement "
 none of them reported severe side effects or required hospitalization. " is just on the level of analysis of survey without non-response technique correction. More should be written, or sub-sentence should be eliminated. 

Addendum to    Major Comment 2: Authors are correct, there was no clinical trial involved in your study.
But how the ethical committee analysed your " online-based survey questionnaire to collect our data"?
This should be carefully and explicitly described in order to avoid ethical issues, or conflicts of interests. 
Authors, based on the journal policy should also enclose Conflict of interest statement
where the   competing interests should be explained (e.g. personal fees  and non-financial support from involved  companies).  
Moreover, since Referee is active in Ethical Comites on clinical trials, some appropriate model of   clinical trial model which can be compatible with your
online-based survey questionnaire to collect your data will be important to introduced, including standard  discussion on tuning of the parameters.
If this was a cross-sectional prospective study based on the data collected from random participants who had been previously vaccinated, how, for instance frauds in filling of formulary are assessed? Authors did not vaccinate the participants for the both the study groups.  

    Answer to    Major Comment 3 is satisfactory.

Addendum to    Major Comment 4:
The answer is still not satisfactory. this is a very important point. Please, comment also on absolute risk reduction and properly cite 
doi: 10.1016/S2666-5247(21)00069-0

    Addendum to        Major Comment 5: As authors correctly stated and they agreed that results from mice are not 100% projectile to humans and need for clinical trial assessment of heterologous prime-boost regimens. Therefore in order to avoid long discussion, eliminate reference to the data of mice.
This imprecision seriously corrupts your manuscript.  

Addendum to        Major Comment 6:  your manuscript relates to mixed therapies. If you do not have published reference for the  results that mixed strategy
is higher humoral immune response but also a cellular immune response, please, avoid this part.
This imprecision seriously corrupts your manuscript.

Addendum to        Major Comment 7: As authors correctly stated and they agreed that reporting bias occurs, they need to make some simple assessing of its numerical value in their data setup. More discussion on this is necessary and needed. 

Addendum to Major Comment 8:
Unfortunately, authors did not put attention to this point. The statement in Section 2 ". To perform statistical analysis, the chi-square test was used. We conducted a statistical analysis of the 136
data using SPSS v.23 (IBM Corp, Armonk, NY, USA), with a p <= 0.05 as significant."
is unfortunately incorrect, since  Application of chi^2 test is not justified. 
Authors are correct that "random sampling includes ease of use and accuracy of representation", however and unfortunately, no random sample is used in this survey, since   sample is not at random. This is a sort of Convenience sampling. Authors can read about such sampling which is not at random in books, e.g. in Mark R. Learly, Behavioral Research Methods, 2nd edition or in any specialised book on Non-random Sampling.  
 One of the assumptions of the Chi-square is that sampling method is simple random sampling. thus, using of p-value cut-off 0.05 does not holds its nominal significance level. Statistics should be improved, chi^2 test in its classical form is not reliable. Thus, several conclusions are unjustified from the obtained p-values.
This should be checked and explained in a proper way. 

Author Response

Reviewer 1

Addendum to Major Comment 1: Sample size is extremely small:
It should be also clearly written in 5. Conclusion, so you cannot (based of very low sample size and non-comparable sample sizes) guarantee that a mix-vaccine strategy is safe. Moreover, there is no objective justification of the sentence "Therefore, based on our current study, a mix-vaccine strategy is an effective way to compromise the shortages of COVID-19 vaccines and to accelerate the vaccination pro- 296 cess globally." and it should be removed. You should be careful that non-reporting effects these types of survey, and your statement " none of them reported severe side effects or required hospitalization. " is just on the level of analysis of survey without non-response technique correction. More should be written, or sub-sentence should be eliminated.

The authors thank the reviewer for the addendum to comment 1. We have removed the sentences that the learned reviewer suggested from the conclusion and we also added the sample size in the conclusion and re-wrote the conclusion (line 409-416). 

Addendum to Major Comment 2: Authors are correct, there was no clinical trial involved in your study.
But how the ethical committee analysed your " online-based survey questionnaire to collect our data"?
This should be carefully and explicitly described in order to avoid ethical issues, or conflicts of interests. 
Authors based on the journal policy should also enclose Conflict of interest statement where the competing interests should be explained (e.g. personal fees and non-financial support from involved companies). Moreover, since Referee is active in Ethical Comites on clinical trials, some appropriate model of clinical trial model which can be compatible with your online-based survey questionnaire to collect your data will be important to introduced, including standard discussion on tuning of the parameters. If this was a cross-sectional prospective study based on the data collected from random participants who had been previously vaccinated, how, for instance frauds in filling of formulary are assessed? Authors did not vaccinate the participants for the both the study groups. 

The authors thank the reviewer for this comment. Authors acknowledged during data collection with an ethical obligation to respect each participant's autonomy. Cross-sectional, retrospective study using an online questionnaire was conducted confidentiality and informed consent was provided. Legal obligations for data protection were followed, as must the respondent's right to secrecy. Participants were thoroughly informed of the survey's aims and the participants' consent to participate in the survey was obtained and recorded. This was added to the revised manuscript (line 423-429). Conflict of interest statement was also added to the revised manuscript (line 431-432). Funding statement was also added to the revised manuscript (line 420-422). Moreover, according to the suggestion of the learned reviewer, a clinical trial model was added to the limitation section in the revised manuscript (line 402-406).

Answer to Major Comment 3 is satisfactory.

We are grateful to the learned reviewer for the appreciation.

Addendum to Major Comment 4:
The answer is still not satisfactory. this is a very important point. Please, comment also on absolute risk reduction and properly cite 
doi: 10.1016/S2666-5247(21)00069-0

We are thankful to the learned reviewer for raising this point. We worked on this point raised by the learned reviewer and have amended the sentence commenting on the absolute risk reduction and cite the suggested paper by the learned reviewer (line 115-122).

Addendum to Major Comment 5: As authors correctly stated and they agreed that results from mice are not 100% projectile to humans and need for clinical trial assessment of heterologous prime-boost regimens. Therefore, in order to avoid long discussion, eliminate reference to the data of mice.
This imprecision seriously corrupts your manuscript.

We totally agree with the learned reviewers that data of mice would impact our manuscript. We are thankful to the learned reviewer for raising this point and we removed the data of mice in the revised manuscript.  

Addendum to Major Comment 6:  your manuscript relates to mixed therapies. If you do not have published reference for the results that mixed strategy is higher humoral immune response but also a cellular immune response, please, avoid this part. This imprecision seriously corrupts your manuscript.

 We are thankful to the learned reviewer for raising this point also and we removed the data related to the immunological response according to the reviewer’s suggestion in the revised manuscript.  

Addendum to Major Comment 7: As authors correctly stated and they agreed that reporting bias occurs, they need to make some simple assessing of its numerical value in their data setup. More discussion on this is necessary and needed. 

Thank you for this comment. Multivariate analyses were performed and added to the revised manuscript (line 217-218), (line 279-293).  

Addendum to Major Comment 8:
Unfortunately, authors did not put attention to this point. The statement in Section 2 ". To perform statistical analysis, the chi-square test was used. We conducted a statistical analysis of the 136 data using SPSS v.23 (IBM Corp, Armonk, NY, USA), with a p <= 0.05 as significant." is unfortunately incorrect, since Application of chi^2 test is not justified.  Authors are correct that "random sampling includes ease of use and accuracy of representation", however and unfortunately, no random sample is used in this survey, since   sample is not at random. This is a sort of Convenience sampling. Authors can read about such sampling which is not at random in books, e.g. in Mark R. Learly, Behavioral Research Methods, 2nd edition or in any specialised book on Non-random Sampling. One of the assumptions of the Chi-square is that sampling method is simple random sampling. thus, using of p-value cut-off 0.05 does not holds its nominal significance level. Statistics should be improved, chi^2 test in its classical form is not reliable. Thus, several conclusions are unjustified from the obtained p-values. This should be checked and explained in a proper way. 

The authors thank the learned reviewer for this valuable comment. Our convenience sampling may limit the possibility to generalize our findings to a wider population, however our target sample was hard to reach, because the only way to reach participants was through social media, therefore, we were not able to apply random sampling in this study. To the best of our knowledge, convenience sampling does not have any restrictions for the statistical tools to be used Therefore, in our study, the chi-square test was used [References have been inserted in the bibliography of the revised manuscript 16-22] to check whether the differences between groups are statistically significant or not. This has been added to the revised manuscript (line 238-246) and (line 373-379).

Reviewer 2 Report

There were seeveral major flaws in study design limiting the values of this study. Athough the authors addressed these limitations, I would like to suggest the authors develop a new study to examining the same study questions.

Author Response

Reviewer 2 (Round 2) comment: There were several major flaws in study design limiting the values of this study. Although the authors addressed these limitations; I would like to suggest the authors develop a new study to examining the same study questions.

The authors are grateful to the reviewer for this comment on our paper. We are really appreciating the learned reviewer for this point to develop a new study by examining the same questions as the reviewer suggested. However, we are unable to repeat this study from scratch as Oxford-AstraZeneca vaccine discontinued in Saudi Arabia and the only available vaccines now are Pfizer-BioNTech and Moderna. Moreover, most of population in Saudi Arabia have get vaccinated long time ago, so we will be unsure if individuals are still remembering the side effect they had after vaccination. Furthermore, it will be difficult to reach individuals who mixed vaccine doses as there is not any available vaccines in Saudi Arabia now except Pfizer-BioNTech and Moderna. Most people are usually getting the same vaccine for both doses for unknown reason so that is why it is difficult to target mixed group. Finally, the statistics of individuals in Saudi Arabia shows that most people started taking the third dose, therefore, this will not be appropriate for our study as we are targeting two doses only. Given that, developing a new study at the meantime would be beyond the scope of our study.

Round 3

Reviewer 1 Report

Authors again did a job in order to answer my major comments, but still   more work is needed. Practically, they succeeded to reply completely to Addendum to    Major Comment 1,2,4,5 &6:.
The other major comments 7 & 8 need  more attention and work, according to the following queries, which can help Authors to better address the original Major comments.

Addendum to        Major Comment 7:  Authors added  Multivariate analyses   (line 217-218), (line 279-293).
However, more description of these analyses is needed. E.g. which analyses, how these were performed, etc. 

Addendum to Major Comment 8: Authors correctly addressed that the do  convenience sampling. However, it is not true that convenience sampling is not having any theoretical regularities before application of   formal statistical tests, i.e. chi^2 test in its classical form, see McHugh ML. The chi-square test of independence. Biochem Med (Zagreb). 2013;23(2):143-149. doi:10.11613/bm.2013.018
This need serious remake, since inappropriate application of chi^2 test lead to wrong results. 
This should be checked and explained in a proper way.

Author Response

Please find the response in the attached file. 

Reviewer 2 Report

i have no further comment.

Author Response

(The authors gave the same response as above.)
